# Post-Transcriptional Effects of miRNAs on PCSK7 Expression and Function: miR-125a-5p, miR-143-3p, and miR-409-3p as Negative Regulators

**DOI:** 10.3390/metabo12070588

**Published:** 2022-06-23

**Authors:** Mahshid Malakootian, Parisa Naeli, Seyed Javad Mowla, Nabil G. Seidah

**Affiliations:** 1Laboratory of Biochemical Neuroendocrinology, Clinical Research Institute of Montreal, Montreal, QC H2W1R7, Canada; mahshid.malakootian@gmail.com; 2Cardiogenetic Research Center, Rajaie Cardiovascular Medical and Research Center, Iran University of Medical Sciences, Tehran 1995614331, Iran; 3Molecular Genetics Department, Faculty of Biological Sciences, Tarbiat Modares University, Tehran 14115-111, Iran; p.naeli@qub.ac.uk (P.N.); sjmowla@yahoo.com (S.J.M.); 4Centre for Cancer Research & Cell Biology, Queen's University Belfast, Belfast BT9 7AE, UK

**Keywords:** *PCSK7*, miR-125a-5p, miR-143-3p, miR-409-3p, RNA regulator, proprotein convertases

## Abstract

The regulatory mechanism of *PCSK7* gene is still unknown, although its encoded protein PC7 is the most ancient and highly conserved of all proprotein convertases and exhibits enzymatic and non-enzymatic functions in liver triglyceride regulation. Bioinformatics algorithms were used to predict regulatory microRNAs (miRNAs) of *PCSK7* expression. This led to the identification of four miRNAs, namely miR-125a-5p, miR-143-3p, miR-409-3p, and miR-320a-3p, with potential binding sites on the 3′-untranslated region (3′-UTR) of human *PCSK7* mRNA. The expression patterns of these miRNAs and *PCSK7* mRNA were assessed in three different cell lines with quantitative polymerase chain reaction (qPCR), which revealed reciprocal expression patterns between the expression levels of the four selected miRNAs and *PCSK7*. Next, the interactions and effects of these miRNAs on *PCSK7* expression levels were investigated via cell-based expression analysis, dual-luciferase assay, and Western blot analysis. The data revealed that *PCSK7* mRNA levels decreased in cells transfected with vectors overexpressing miR-125a-5p, miR-143-3p, and miR-409-3p, but not miR-320a-3p. The dual-luciferase assay demonstrated that the above three miRNAs could directly interact with putative target sites in *PCSK7* 3′-UTR and regulate its expression, whereas miR-320-3p exhibited no interaction. Western blot analysis further revealed that the overexpression of miR-125a-5p in Huh7 cells inhibits the expression and ability of PC7 to cleave human transferrin receptor 1. Our results support a regulatory role of these miRNAs on *PCSK7* expression and function and open the way to assess their roles in the regulation of PC7 activity in vivo in the development of hepatic steatosis.

## 1. Introduction

Proprotein convertases (PCs) comprise a family of nine secretory proteases related to bacterial subtilisin and yeast kexin (genes *PCSK1* to *PCSK9*) [1]. PCs cleave their protein precursors at basic and non-basic amino acid sites. Seven out of nine PCs, namely PC1/3, PC2, Furin, PC4, PC5/6, PACE4, and PC7, belong to a subgroup that cleaves its substrates at either single or pairs of basic amino acids within the motif [R/K]-2(Xn)-[R/K]↓ (Xn stands for 0–3 spacer amino acids), while SKI-1/S1P and PCSK9 cleave their substrates and/or themselves at non-basic sites [1].

The *PCSK7* gene, harbored on chromosome 11q23.3, encodes the most ancient and highly conserved type-I membrane-bound protease, termed “PC7”, which is ubiquitously expressed [2]. The mature form of PC7, as is the case with all other PCs, is the result of a sequential two autocatalytic processing (cleavage) events. PC7 is initially synthesized as a proPC7 zymogen (~102 kDa) in the endoplasmic reticulum (ER), which is then autocatalytically cleaved in the ER to produce an inactive PC7 (~92 kDa) that is noncovalently bound to its N-terminal inhibitory prodomain [3,4]. Further, posttranslational modifications of PC7 such as N-glycosylation, Tyr-sulfation, and Cys-palmitoylation of the cytosolic tail at residues Cys_699_ and Cys_704_ occur in the ER and *cis-medial* -Golgi [2,5,6]. The prodomain-PC7 complex is then activated in endosomes and constantly transits between the *trans*-Golgi network and the cell surface [7,8,9,10,11].

The various mechanisms underlying PC7 activation and functions are still not well defined; nonetheless, it is now known that PC7 exerts its activity via enzymatic [8,12] and non-enzymatic [13] mechanisms to regulate specific protein targets. As a protease, PC7 is specifically implicated in iron homeostasis by shedding human transferrin receptor 1 (hTfR1). To date, the shedding of hTfR1 is the only recognized specific substrate of PC7 cleaved at Arg_100_↓ within the sequence KTECE**R**_100_↓LA [8]. PC7 is also implicated in the shedding of the cancer-associated proteins CASC4 and GPP130 [14]. We also demonstrated that PC7 enhances the processing of the cell surface pro-epidermal growth factor [15] and brain-derived neurotrophic factor (BDNF) [12]. In addition, triglyceride metabolism is modulated by PC7, in part owing to its ability to non-enzymatically regulate the levels of circulating apolipoprotein A-V [13], an activator of lipoprotein lipase, and possibly other secretory lipoproteins starting in the endoplasmic reticulum. This led to the suggestion that PC7 inhibition could be useful to treat metabolic dysfunction-associated fatty liver disease (MAFLD) and hepatic steatosis [16,17], and possibly in iron regulation in hereditary haemochromatosis [18]. In fact, it has been suggested that PC7 could be a target for the treatment of dyslipidemias, e.g., triglyceride-associated fatty liver disease [19].

Accumulating research in recent years has demonstrated that microRNAs (miRNAs), which constitute a highly evolutionarily conserved class of small (18–22 nucleotides) noncoding RNAs, post-transcriptionally modulate gene expression by either repressing translation or promoting messenger RNA (mRNA) degradation via binding to the three prime untranslated regions (3′-UTRs) of their target mRNAs [20,21,22]. Acting as fine-tuners or micromanagers of gene expression of more than 60% of human protein-coding genes, these small-molecule regulators effectively control a wide spectrum of molecular pathways during embryonic development and in disease states [23,24,25].

Only a handful of studies have demonstrated a regulatory role of miRNAs in controlling the expression of PCs [20,26]. For instance, miR-124 directly represses the expression of PACE4 and reduces the cell growth and cell invasion of prostate cancer [27]. In addition, miR-24 is a Furin-directed post-transcriptional regulator that modulates the Furin-mediated activation of the hemagglutinin precursor and the production of fusion-competent virions in the host’s secretory pathway [28]. Our group reported direct interactions between miR-191, miR-222, and miR-224 and the 3′-UTR of *PCSK9* mRNA, leading to the inhibition of *PCSK9* expression [20]. In addition, recent high-throughput screens identified six other miRNAs that reduce PCSK9 levels, including miR-221-5p, miR-342-5p, miR-363-5p, miR-609, miR-765, and miR-3165 [26]. However, no report has yet appeared on the regulation of PC7 levels by miRNAs.

Even though PC7 is highly conserved among PCs, the molecular post-transcriptional regulatory mechanisms of *PCSK7* are unknown. In the present study, we drew upon bioinformatics to predict miRNAs that target the 3′-UTR of human *PCSK7* mRNA and then experimentally validated the effects of the predicted miRNAs on the expression and function of *PCSK7*. We uncovered that miR-125a-5p, miR-143-3p, and miR-409-3p could repress *PCSK7* expression post-transcriptionally by targeting its 3′-UTR. Our findings indicate a regulatory network between miRNAs and *PCSK7* expression and suggest that the three above-mentioned miRNAs are novel modulators of the expression of *PCSK7*. This information may lead to novel approaches to silence PCSK7 expression to mitigate the development of fatty liver disease and its associated sequelae.

## 2. Results

### 2.1. Bioinformatics Prediction of PCSK7-Targeting miRNAs

The molecular mechanisms underlying the regulation of *PCSK7* expression by miRNAs were investigated via a bioinformatics analysis of five different publicly available target prediction programs, namely TargetScan, DIANA-micro-T, miRDB, miRanda, and UCSC, to predict miRNAs targeting the 3′-UTR of the *PCSK7* transcript. Several miRNAs were identified by these programs, leading to our selection for further analysis of four different targeting miRNAs commonly predicted by all programs, namely miR-125a-5p, miR-143-3p, miR-409-3P, and miR-320a-3p. Among the chosen miRNAs, miR-125a-5p had four target sites, including two in the 3′-UTR and the rest on exons 14 (E14) and 15 (E15) of *PCSK7,* and miR-143-3p has two target sites within the 3′-UTR of *PCSK7*. In contrast, miR-409-3P and miR-320a-3p each have only one predicted target site (Figure 1).

### 2.2. Expression of the Predicted miRNAs and PCSK7 in HEK293T, HepG2, and Huh7 Cells

We next used qPCR to evaluate the endogenous expression patterns of the four selected miRNAs (miR-125a-5p, miR-143-3p, miR-409-3p, and miR-320a-3p) and *PCSK7* mRNA in three cell lines, namely kidney-derived HEK293T, hepatocytes-derived HepG2, and Huh7 cells (Figure 2).

Huh7 cells showed the highest relative expression of *PCSK7* mRNA and extremely low RNA expression levels of miR-143-3p and miR-409-3p. The expression levels of miR-125a-5p and miR-320a-3p were similar in all cell lines. In HepG2 and HEK293T cells, the expression levels of miR-125a-5p and miR-320-3p were comparable to that of *PCSK7*, while the expression levels of miR-143-3p and miR-409-3p were low. From these data, it seems that the levels of *PCSK7* mRNA are inversely correlated to the RNA levels of miR-143-3p and miR-409-3p in all cells, whereas miR-125a-5p may negatively regulate *PCSK7* mRNA levels in Huh7 cells.

### 2.3. Negative Regulation of PCSK7 mRNA Expression Levels by the Overexpression of miR-125a-5p, miR-143-3p, and miR-409-3p

Huh7 cells were transfected with miR-overexpressing vectors carrying the precursors of miRNA sequences and mock-related counterpart vectors to assess the possible relationship between the expression patterns of miR-125a-5p, miR-143-3p, miR-409-3p, and miR-320a-3p with *PCSK7* expression at the transcriptional level. The qPCR results obtained at 48 h following transfection demonstrated that the overexpression of miR-143-3p and miR-409-3p in Huh7 cells significantly decreased the expression of *PCSK7* mRNA by 40–50% compared to cells transfected with the mock control vector (*p* = 0.0158 and *p* = 0.0084) (Figure 3A). Although the overexpression of miR-125a-5p tended to reduce the expression of *PCSK7* mRNA in these cells, this decrease did not reach statistical significance (*p* = 0.0693). In contrast, the ectopic expression of miR-320a-3p did not downregulate the expression of *PCSK7* in Huh7 cells (*p* = 0.5538).

The association between the expression levels of miR-125a-5p and miR-320a-3p and the expression level of *PCSK7* was further investigated by measuring the levels of *PCSK7* mRNAs in two other cell lines, namely HEK293T and HepG2 cells. The qPCR results showed that the overexpression of miR-125a-5p now led to a significant ~25% reduction in *PCSK7* mRNA levels 48 h after transfection in these cells (*p* = 0.0114, *p* = 0.0010, respectively) (Figure 3B and Appendix A). However, like Huh7 cells, no significant regulatory relationship was observed for miR-320a-3p in HEK293T (*p* = 0.9972) and HepG2 (*p* = 0.7830) cells. We conclude that the overexpression of miR-143-3p, miR-409-3p, and less so miR-125a-5p may downregulate *PCSK7* expression.

### 2.4. Direct Interaction between the Predicted miRNAs and the 3′-UTR of PCSK7

A dual-luciferase assay was applied to investigate the interaction between the predicted miRNAs and the 3′-UTR of human *PCSK7* mRNAs, cloned downstream of the *Renilla luciferase* gene in the psiCHEK-2 plasmid. Additionally, miR-224-5p without any target site on the 3′-UTR of *PCSK7* served as a control. Here, the overexpression of miR-125a-5p significantly reduced the relative luciferase activity by ~50% (*p* < 0.0001) in HEK293T cells co-transfected with the miR-125a-5p-overexpressing plasmid and a psiCHECK-2 carrying the wild-type *PCSK7* 3′-UTR (Figure 4). Likewise, miR-143-3p and miR-409-3p significantly decreased the luciferase activity by ~20-30% by targeting the *PCSK7* 3′-UTR (*p* = 0.0013 and *p* = 0.04, respectively). However, miR-320a-3p and miR-224-5p, used as controls, had no significant effect (*p* ≥ 0.05 and *p* > 0.99) on luciferase activity. Accordingly, miR-125a-5p, miR-143-3p, and miR-409-3p were chosen for further analysis of their possible direct interactions with the *PCSK7* 3′-UTR.

Another set of luciferase assays in HEK293T and HepG2 cells were performed to confirm the interaction of miR-125a-5p (Appendix A). The 3′-UTR (whole is 926 bp) of the *PCSK7* were divided into two parts, the proximal segment (647 bp, UTR-1) and the distal segment (275 bp, UTR-2), and separately cloned downstream of the *Renilla luciferase* gene in the psiCHEK-2 plasmid (Appendix A). The experiments were performed in three biological replicates. As indicated in Figure 1, miR-125a-5p has one target site in each segment. Briefly, each miR-overexpressing vector was co-transfected with each UTR constructs (Wildtype 3′-UTR, 3′-UTR-1, and 3′-UTR-2) in HEK293T and HepG2 cells. Forty-eight hours after transfection, the dual-luciferase assay was applied. Similar effects on *PCSK7* UTRs were observed in both cell lines (Appendix A). Here, the overexpression of miR-125a-5p significantly reduced the relative luciferase activity in the HEK293T cells and HepG2 cells, co-transfected with the miR-125a-5p-overexpressing plasmid and a psiCHECK-2 carrying the wild-type *PCSK7* 3′-UTR, the first part of *PCSK7* 3′-UTR (UTR-1), and the second part of *PCSK7* 3′-UTR (UTR-2). Likewise, miR-125a-5p targeted both UTR-1 and UTR-2 and the wild-type *PCSK7* 3′-UTR in both transfected cell lines: in HEK293T cells (*p*-values were *p* < 0.0001, *p* = 0.0043, and *p* < 0.0001, respectively) and in HepG2 cells (*p* < 0.0001, *p* = 0.0247, and *p* < 0.0001, respectively). Both UTR-1 and UTR-2 harbored one target site of miR-125a-5p (Figure 1).

The effects of the combination of predicted microRNAs (mix microRNAs) on PC7 expression were assessed via co-transfection of the combined the miRNAs and 3′-UTR of the *PCSK7.* However, the fold change in the luciferase activity did not vary significantly compared to the effect of each individual microRNAs (Appendix A). This could be due to several factors such as the accessibility of different miRNA target sites in the cells, i.e., when one miRNA targets the mRNA, the site for another miRNA might not be accessible. Therefore, we omitted the mix microRNA analyses for further analysis.

### 2.5. Direct Interactions between miR-125a-5p, miR-143-3p, and miR-409-3p with the 3′-UTR of PCSK7

A luciferase reporter assay further confirmed that miR-125a-5p, miR-143-3p, and miR-409-3p directly targeted the 3′-UTR of the *PCSK7* transcript since the negative regulatory effect was lost upon the deletion of the miRNA target sites (miR-125a-5p, miR-143-3p, and miR-409-3p) on the 3′-UTR of the *PCSK7* reporter plasmid (Figure 5). In this regard, three different mutated miR plasmids were constructed for both miR-125a-5p and miR-143-3p. Single-target sites of each miRNA were omitted individually in two separate plasmids, and the two putative binding sites of the miRNAs were deleted in the other plasmid. The data demonstrate that overexpression of miR-125a-5p still decreased the relative luciferase activity in the HEK293T cells co-transfected with the plasmids carrying a mutated form of *PCSK7* 3′-UTR, containing a single deletion of miR-binding sites (Mut-a and Mut-b), compared with cells co-transfected with the mock counterpart vectors along with the same mutant vectors (Mut-a and Mut-b) (*p* = 0.04 and *p* = 0.01). In contrast, a mutated form of *PCSK7* 3′-UTR, featuring the deletion of both miRNA target sites (Mut-a,b), failed to significantly decrease the relative luciferase activity in these cells (*p* = 0.24) (Figure 5A). As was observed for miRNA-125a-5p, the overexpression of miR-143-3p had no significant effect on luciferase activity when the two putative target sites (Mut-a,b) were omitted from the 3′-UTR of *PCSK7* in HEK293T cells (*p* = 0.98). By comparison, luciferase activity was considerably diminished in these cells expressing the miR-143-3p plasmid and mutated vectors carrying only one miRNA response element sequence (Mut-a; *p* = 0.03 and Mut-b; *p* = 0.004) (Figure 5B). Finally, miR-409-3p overexpression did not lead to a reduction in luciferase activity when the only putative target site of miR-409-3p was absent from the 3′-UTR of *PCSK7* (*p* = 0.40) (Figure 5C).

Overall, the above results confirm the presence of two independent miRNA response elements for both miR-125a-5p and miR-143-3p and one miRNA response element for miR-409-3p as active elements. We conclude that miR-125a-5p, miR-143-3p, and miR-409-3p could regulate the expression of *PCSK7* mRNA through direct interactions with these target miRNA-binding sites.

### 2.6. Functional Effects of miR-125a-5p on PC7 Activity

Western blot analysis was conducted to determine whether miR-125a-5p affected the functional activity of endogenous PC7 in Huh7 cells. A wild-type human PC7-overexpressing vector was co-transfected with pre-miR-125a-5p and pre-miR-224-5p-overexpressing plasmids in Huh7 cells. Western blot results at 48 h post-transfection revealed that overexpressed PC7 protein levels fell significantly by ~80% (*p* = 0.0025) following miR-125a-5p co-expression, compared with the mock-related (pEGFP-C1) and miR-224-5p controls (Figure 6A,B). As a control, a human PCSK9-overexpressing vector was co-transfected with miR-125a-5p and miR-224-5p vectors in Huh7 cells, and these miRNAs had no effect on the expression level of PCSK9 protein (not shown).

Next, Huh7 cells were co-transfected with vectors expressing PC7 and miR-125a-5p at DNA ratios of 1:2 (2X), 1:3 (3X), and 1:5 (5X). The results showed that all three ratios of miR-125a-5p significantly decreased PC7 protein levels by >80%, with no significant PC7-silencing differences between the different amounts of miR-125a-5p (Appendix A).

We previously reported that PC7 specifically cleaves the human type-II membrane-bound hTfR1 into a soluble secreted form, thereby enhancing its circulating levels [8]. In addition, our bioinformatics analysis based on the UCSC genome browser (miRcode predicted microRNA target sites track) demonstrated that miR-125a-dp does not have any target sites on the 3′-UTR of hTfR1. In agreement with this fact, miR-125a-5p that reduced the level of the PC7 protein (Figure 6, Appendix A) exerted no effect on the expression level of the endogenous hTfR1 protein in Huh7 cells (Appendix A).

Accordingly, for the assessment of the functional effect of miR-125a-5p on PC7 activity, hTfR1-V5 was co-expressed with PC7 and different amounts of the miR-125a-5p (2X, 3X, and 5X)-expressing vectors in the Huh7 cells (Figure 7). The data show that the presence of miR-125a-5p led to a decline in the ability of PC7 to shed hTfR1 into the media (Figure 7). These results demonstrate that miR-125a-5p not only diminished the expression level of *PCSK7* mRNA but also functionally abrogated its protease activity on hTfR1.

## 3. Discussion

PCs constitute a family of nine members. With their irreversible limited proteolysis function, they play key roles in regulating both physiological and pathophysiological conditions by activating or inactivating a wide spectrum of precursor proteins, including growth factors, hormones, receptors, and adhesion molecules, thereby generating cleaved products with novel functions [1,13,29]. Despite all accumulating evidence on PCs, the significance, regulation, and specific functions of PC7, their most conserved and ancient family member, have yet to be fully elucidated.

Nonetheless, the ubiquitously expressed PC7 has been implicated in iron metabolism [8], anxiety/mood regulation [12], triglyceride metabolism [30,31], and breast cancer progression [14]. On the other hand, miRNAs have pivotal roles in controlling such physiological processes as development, differentiation, apoptosis, proliferation, and metabolism, and dysregulated miRNAs are associated with various pathological conditions [23]. In the present study, we unveiled miR-125a-5p, miR-143-3p, and miR-409-3p as natural negative regulators of *PCSK7* mRNA via bioinformatics and *in cellulo* functional enzymatic activity of human PC7. We found that miR-125a-5p, miR-143-3p, and miR-409-3p directly targeted *PCSK7* mRNA and downregulated its expression, whereas miR-320a and miR-224 did not target *PCSK7*. Additionally, miR-125a-5p not only suppressed the expression of PC7 protein but also reduced its protease function on hTfR1. Our study is the first investigation to report regulatory effects of miRNAs on *PCSK7* expression and function. We note, however, that such regulation of *PCSK7* must be viewed in the context that each miRNA may target multiple mRNAs and thus regulate complex cellular pathways.

Research in the last decade has established that miRNAs, with their highly organ- and cell-specific expression patterns, participate in the modulation of the expression of targeted genes through transcription regulation, mRNA degradation, translation inhibition, and translation activation [20,32,33,34]. Our expression analysis validated the differential expression of miRNAs and *PCSK7* in different cell lines.

PC7 has been proposed to play critical roles in liver triglyceride regulation, including non-alcoholic fatty liver disease (NAFLD) and hepatocellular carcinoma (HCC) [13,17,35]. Interestingly, miR-125a-5p, miR-143-3p, and miR-409-3p participate in a variety of diseases, including liver diseases and hepatic cancer [36,37,38,39]. Previous studies have demonstrated that miR-125a-5p binds to the viral transcript encoding the surface antigen of the hepatitis B virus and represses viral replication by interfering with the expression of this antigen [40,41]. Likewise, miR-125a-5p in the liver is associated with the replication and progression of the hepatitis B virus in patients with chronic hepatitis B [41]. In addition, serum levels of miR-125 could predict the progression of liver diseases [40]. The differential expression of miR-125-a-5p in the progression of high-fat-diet-induced non-alcoholic fatty liver disease–non-alcoholic steatohepatitis–hepatocellular carcinoma (NAFLD–NASH–HCC) was investigated by Tessitore et al. [42], who reported that miR-125a-5p and miR-182 exhibited early and significant dysregulation in the sequential hepatic damage process [42]. Therefore, the expression pattern of miR-125a-5p is a suitable marker of liver disease. Another miRNA, miR-143-3p, suppresses the proliferation and invasion of HCC cells by regulating the *FGF1* gene [43]. Further, the differential expression of miR-143 is associated with HCC, downregulating the expression levels of the Toll-like receptor 2 (TLR2), nuclear factor-kappa B (NF-κB), and matrix metallopeptidase 2 and 9 (MMP-2 and MMP-9) [44]. Mamdouh et al. concluded that the upregulation of miR-143, miR-215, and miR-224 in the serum of the hepatitis C virus-associated HCC patients compared with control samples could be applied not only as a diagnostic marker of the disease but also as an indicator of the grade of the tumor and the stage of fibrosis, although all three miRNAs were downregulated in liver tumor tissue compared with their control. Conversely, the development of liver fibrosis in autoimmune hepatitis was lessened by miRNA-143-3p through the regulation of TAK1 phosphorylation according to another study [39,45]. Investigations, both *in vivo* and *in vitro*, have suggested that miR-409-3p, along with other miRNAs, plays a crucial role in regulating the progression of human NAFLD and is a non-invasive diagnostic biomarker of its severity and progression [46,47]. In addition, miR-409-3p controls the angiogenesis of brown tissue and insulin resistance by regulating endothelial cell–brown adipose tissue crosstalk via a MAP4K3-ZEB1-PLGF signaling axis [48]. The expression of miR-409-3p is reduced in the tissue and cell lines of liver cancer and is negatively correlated with tumor stage, tumor size, and the survival time of patients [38].

Our luciferase reporter assay validated and confirmed that *PCSK7* mRNA is a direct target of miR-125a-5p, miR-143-3p, and miR-409-3p. Since PC7 is implicated in liver diseases and hepatic cancer, our results suggest that these miRNAs could modulate *PCSK7* expression in different liver diseases, likely being protective against the development of MAFLD/NAFLD [16], where PC7 seems to play a non-enzymatic role [13,17], possibly as a chaperone for some lipoproteins such as apolipoprotein B (Vatsal S. et al., *in preparation*). However, further investigations are needed to specify which miRNAs significantly regulate the expression of *PCSK7* in different liver pathologies and/or cancers.

Cirrhosis develops in <10% of individuals homozygous for the C282Y variant in the homeostatic iron regulator (HFE) gene, where a gain-of-function variant of *PCSK7* (rs236918) [8,49] is associated with an increased risk of cirrhosis in this patient population [18]. The various mechanism regulating PC7 are not yet defined, except that its Ser-phosphorylation can reduce its non-enzymatically induced degradation of apoA-V in the ER [13]. We previously showed that PC7 regulates iron metabolism in the liver by an enzymatic mechanism implicating the shedding hTfR1, resulting in the secretion of a circulating sTfR1 that correlates with iron deficiency [8,49]. Our present Western blot analysis demonstrated that *PCSK7* inhibition by miR-125a-5p reduced the shedding of hTfR1, suggesting that miR-125a-5p could reduce the enzymatic function of PC7 and inhibit the shedding of hTfR1 and could find applications in the treatment of hereditary hemochromatosis.

Overall, we validated our hypothesis that the gene expression of *PCSK7* can be downregulated by miRNAs, three of which were identified in this study. In the future, the expression of mimics of such miRNAs or others would represent a novel approach to decrease the activity of PC7 and reduce the processing of its downstream substrates, e.g., TfR1, or partners, such as apoA-V and apoB, with pharmacological applications in the treatment of fatty liver disease and dyslipidemia.

## 4. Materials and Methods

### 4.1. Computational Prediction of miRNAs That Target the PCSK7 Gene

The following publicly available bioinformatics algorithms were employed to predict potential miRNAs that target the *PCSK7* gene: TargetScan (http://www.targetscan.org/vert_71/1, accessed on March 2020 and 15 April 2022), DIANA tools (http://diana.imis.athena-innovation.gr/DianaTools/index.php, 15 March 2020), miRDB (http://mirdb.org/,1, accessed on 6 June 2022), miRanda (http://www.microrna.org/microrna/home.do, 1 April 2020), and the UCSC website (http://genome.ucsc.edu, accessed on 15 February 2020 and 15 April 2022).

### 4.2. Cell Culture

Human embryonic kidney 293 (HEK293T), hepatocellular carcinoma (Huh7), and hepatoblastoma (HepG2) cells were selected for further functional experiments. The HEK293T and Huh7 cell lines were cultured in Gibco Dulbecco’s Modified Eagle Medium (DMEM) (Invitrogen, Headquarters Thermo Fisher Scientific, Waltham, MA USA), supplemented with 100 U/mL of penicillin, 100 μg/mL of streptomycin (Sigma, Massachusetts MilliporeSigma, Burlington, MA USA), and 10% fetal bovine serum (FBS) (Invitrogen, Headquarters Thermo Fisher Scientific, Waltham, MA USA); then, they were incubated at 37 °C with 5% CO_2_. The HepG2 cells were cultivated in DMEM-F12 (Invitrogen, Headquarters Thermo Fisher Scientific, Waltham, MA USA), containing 10% FBS, 100 U/mL of penicillin, and 100 μg/mL of streptomycin.

### 4.3. RNA Extraction, Complementary DNA (cDNA) Synthesis, and Quantitative Polymerase Chain Reaction (qPCR)

RNA was isolated from the cells with the TRIzol Reagent (Invitrogen, Waltham, MA, USA) according to the manufacturer’s instructions. The RNA samples were treated with RNase-free DNase I (Fermentas, Waltham, MA, USA) to eliminate any possible DNA contamination. Reverse transcription was performed using the PrimeScript First Strand cDNA Synthesis Kit (Takara, Kusatsu, Japan) following the manufacturer’s protocol. Briefly, one unit of the DNase I enzyme, 1 μL of a buffer, and 1 μg of total RNA were incubated for 30 min at 37 °C. Then, 1 μL of 50 mM EDTA was added for enzyme inactivation and incubated at 65 °C for 10 min. Subsequently, 5 μL of DNase-treated RNA was added to the mix of 0.5 μL of the reverse-transcriptase enzyme, 2 μL of the reverse-transcriptase buffer, and 1 μL of a random hexamer and incubated for 15 min at 37 °C, followed by 5 s at 85 °C for enzyme inactivation. The stem-loop RT-qPCR method was applied for evaluating the expression of miRNAs [50].

The qPCR test was conducted in 20 μL of the PCR reaction mixture using SYBR Green I (Takara, Kusatsu, Japan) in an Applied Biosystems StepOne instrument (Applied Biosystems, Waltham, MA, USA). Briefly, cDNA equivalent to 50 ng of RNA was added to the mix of 10 μL of SYBR Green, 0.5 μM of each primer, 0.4 μL of the ROX reference dye, and sufficient water. The real-time thermal program was as follows: 95 °C for 30 s, 40 cycles at 95 °C for 20 s, and 60 °C for 35 s for *PCSK7*, as well as 95 °C for 30 s, 40 cycles at 95 °C for 5 s, 60 °C for 30 s, and 72 °C for 15 s for miR-125a-5p, miR-143-3p, miR-409-3p, miR-320a-3p, and miR-244, for which RNU48 small nuclear RNA, *β2-microglobulin (β2m)*, and *GAPDH* mRNAs were used as internal controls.

All the reactions were repeated in duplicates. Next, melt curves were analyzed, with the mean threshold cycles used for further analyses. The relative expressions of the miRNAs and *PCSK7* to RNU48 small nuclear RNA and/or *GAPDH* and *β2m* were calculated, respectively, via the 2^−ΔΔCt^ method. All expression experiments were performed in three biological replicates.

The PCR products were sequenced (3500 ABI) to validate the accuracy of the amplification. All the primers for *PCSK7* and the nominated miRNAs are listed in Appendix A.

### 4.4. Plasmids and Cell Transfection

#### 4.4.1. miR-Overexpressing Vectors

The effects of the selected miRNAs on the mRNA expression level of *PCSK7* were examined 48 h after the transfection of the overexpressing miRs in the Huh7 and HEK293T cell lines. Plasmids encoding pEGFP-C-miR-125a-5p, miR-143-3p, miR-409-3p, and miR-320a-3p and their corresponding control (miR-NC) were constructed. The nominated miRNA genes were amplified and cloned downstream of the *GFP* gene into the pEGFP-C1 vector (Clontech, Japan). All the primer sequences used are available in Appendix A.

Pre-miR miRNA precursor-overexpressing vectors (300 ng) were transfected in the Huh7, HEK293T, and HepG2 cell lines using FuGENE HD (Promega Corporation, Madison, WI, USA) in 12-well plates. The transfections were conducted in triplicate, and mock-related counterpart vectors were utilized as controls.

#### 4.4.2. Vectors Containing the PCSK7 3′-UTR Wild-Type and Mutated Forms

The interactions between the miRNAs and their probable targets were explored by cloning the potential target regions in psiCHECK-2 (Promega, USA), a luciferase reporter vector. In psiCHECK-2, *hRluc*, the Renilla luciferase gene, is located upstream of the interest target regions cloned into the psiCHECK-2 vector downstream of the Renilla gene. The region corresponding to the 3′-UTR of *PCSK7* (926-nt sequences in length) that constituted the predicted miRNA response elements was PCR-amplified and cloned downstream of the Renilla luciferase gene in the psiCHECK-2 vector (Promega, USA). For the confirmation of whether miRs response elements on the 3′-UTR of *PCSK7* were active and had direct interactions with the miRNAs, different mutants (plasmids) were constructed via splicing by overhang-extension (SOEing) PCR. For miR-125a-5p and miR-143-3p, each of which has two miRNA response elements on *PCSK7* 3′-UTR, three different mutant constructs were built. Two constructs were made by deleting a putative miRNA target site (about 20 nucleotides), and the third one was made by omitting both putative miRNA target sites (about 40 nucleotides) in the 3′-UTR sequence of *PCSK7*. In the miR-409-3p mutant construct, a putative miRNA target site was deleted from the 3′-UTR sequence. The 3′-UTR of *PCSK7* was divided to two parts: proximal (647 bp, UTR-1) and distal (275 bp, UTR-2). Each part was cloned into the psiCHECK-2 vector downstream of the Renilla gene. (The sequences of the primers are listed in Appendix A).

### 4.5. Luciferase Reporter Assay

The HEK293T cells were co-transfected through the application of the wild-type psiCHECK-2, the mutant *PCSK7* 3′-UTR, and the miR-overexpressing vectors so that the direct interactions of the nominated miRNAs with *PCSK7* 3′-UTR could be investigated. In brief, 150 ng of the wild-type or mutated 3′-UTR constructs and 300 ng of the miRNA-expressing vectors were co-transfected in HEK293T-cultured 48-well plates using FuGENE (Invitrogen, USA). Additionally, the psiCHECK-2 and pEGFP-C1 mock vectors were transfected and utilized as controls for luciferase assay and transfection, respectively. Transfection efficiency was monitored by fluorescent microscopy (Nikon TE2000S, Japan) 36 h following the procedure.

The PsiCHECK-2 reporter construct plasmid contained the *Renilla luciferase* gene upstream of *PCSK7* 3′-UTR and an independent *firefly luciferase* gene as an internal control for normalization. Forty-eight hours after HEK293T co-transfection, the luciferase reporter assay was conducted by employing the Dual-Luciferase Reporter Assay System (Promega, USA) with a luminometer (Titertek-Berthold, Pforzheim, Germany) in accordance with the manufacturer’s protocol. Each sample was performed in triplicate, and the experiment was repeated at least three biological times. In short, a lysis buffer was added to each well after the removal of the media of the cell. Then, LARII Reagent was added, and after 20 min, the firefly luciferase activity was measured as a control. Afterward, Renilla activity was determined using the Stop & Glo Reagent. The relative luciferase activity was calculated using the following formula:ΔFold Activity of Luciferase (Renilla/Firefly) = Average Renilla/Firefly from Samples A/B.

### 4.6. Western Blot Analysis

In the next step, the effect of miR-125a-5p on PC7 function was determined. First, the miR-125a-5p-overexpressing vector and its related mock plasmids were co-transfected with the pIRES vector (Invitrogen, USA), containing the full length of cDNA encoding the *PCSK7* mRNA with the complete 3′-UTR in Huh7 cells. The Western blot analysis was performed 48 h after transfection. Thereafter, the impact of miR-125a-5p on the enzymatic function of PC7 was assessed by co-transfecting Huh7 cells with the miR-125a-5p-overexpressing vector, with that coding for the full length of *PCSK7*, and a plasmid encoding hTfR1 [8,9]. Cell lysates and media were collected for Western blot analysis 48 h after transfection.

Subsequently, proteins were isolated with an ice-cold RIPA buffer (1×), comprising 50 mM of Tris hydrochloride (pH 8), 150 mM of sodium chloride, 0.1% sodium dodecyl sulfate, 1% Nonidet P40, 0.25% sodium deoxycholate, and a cocktail of protease inhibitors (Roche, Oakville, ON, Canada). The proteins were subjected to electrophoresis on 12% polyacrylamide sodium dodecyl sulfate gels and blotted to polyvinylidene fluoride (PVDF) membranes. The PVDF membranes were subsequently blocked by fat-free 5% milk powder dissolved in Tris-buffered saline (0.1 M of Tris hydrochloride (pH 8) and 1.5 M of sodium chloride), containing 0.1% Tween-100 (TBS-T). Both PC7 and TfR1 were C-terminally tagged with V5 and detected with a V5-monoclonal antibody (Invitrogen), and membranes were incubated with appropriate primary and secondary antibodies, as reported [51]. Subsequently, immunoreactive bands (the signal) were visualized with an enhanced chemiluminescent reaction kit (Bio-Rad, Contra Costa County, CA, USA) and recorded via chemiluminescence. The bands were analyzed and quantified using the NIH ImageJ software (US National Institutes of Health, Bethesda, MD, USA).

### 4.7. Statistical Analysis

The 2^−(ΔΔCt)^ method was applied for the qPCR data analysis and gene expression determination. GraphPad Prism version 8 (GraphPad Software, Inc., La Jolla, CA, USA) was employed to analyze the data obtained via the qPCR, dual-luciferase, and Western blot analyses, as well as *p*-value calculation. Student’s t-test and one-way ANOVA statistical tests were applied to analyze the data. A *p* value less than 0.05 was considered statistically significant for all the experiments. The graph bars represent the mean ± SD.

## 5. Conclusions

There is a paucity of information on the regulation of *PCSK7* with noncoding RNAs, especially miRNAs. Herein, we demonstrated that three different miRNAs, namely miR-125a-5p, miR-143-3p, and miR-409-3p, could target and downregulate the expression and function of *PCSK7*. Further investigations are now warranted to determine the details of the underlying mechanism of *PCSK7* regulation by these miRNAs.

## Figures and Tables

**Figure 1 metabolites-12-00588-f001:**
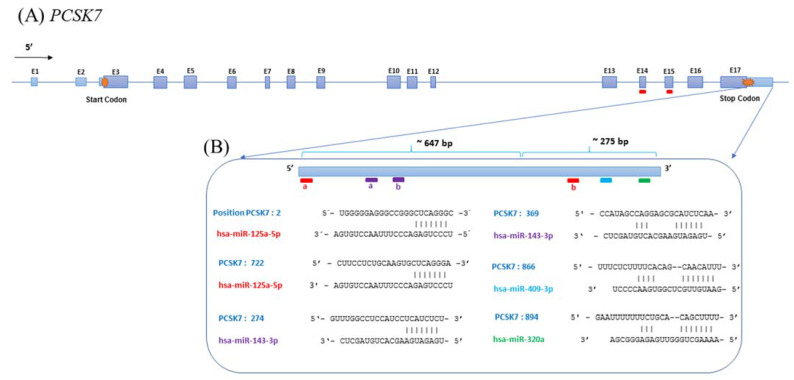
Schematic view of the genome organization of PCSK7. (**A**) Exons, introns, and miRNA target sites on the PCSK7 sequence. Notice the presence of two putative sites targeted by miR-125-5p on exons 14 and 15 (red lines). (**B**) The target sequences of miRNAs in the 3′-UTR of PCSK7 and their nucleotide hybridization status. Note that miR-125-5p (red a and b) and miR-143-3p (violet a and b) target two sites in the 3′-UTR. The target sites of miR-409-3p and miR-320a are presented as blue and green colors, respectively.

**Figure 2 metabolites-12-00588-f002:**
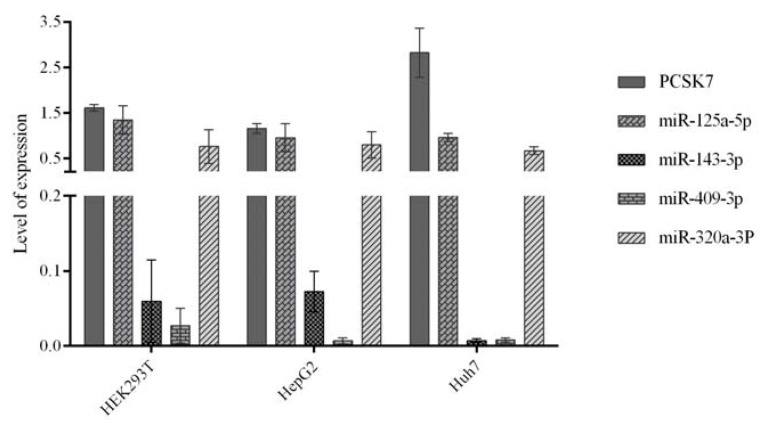
qPCR analysis of the expression levels of *PCSK7* mRNA and predicted microRNAs in three examined cell lines: HEK293T, HepG2, and Huh7. The expression level of *PCSK7* mRNA is the highest in Huh7, whereas miR-143-3p and miR-409-3p have the lowest expression in all the examined cell lines.

**Figure 3 metabolites-12-00588-f003:**
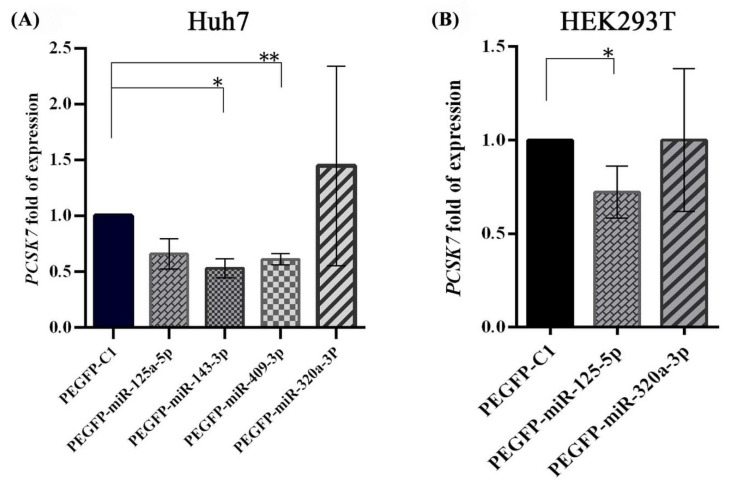
Expression levels of *PCSK7* after miRNA overexpression in Huh7 (**A**) and HEK293T (**B**) cells compared with their mock vector counterpart. A,B) The expression level of *PCSK7* mRNA in Huh7 cells was significantly downregulated by miR-143-3p and miR-409-3p. In both Huh7 and HEK293T cell lines, miR-125a-5p downregulated *PCSK7* expression, although the difference did not reach statistical significance in Huh7 cells (*p* = 0.06). In addition, miR-320a-3p could not downregulate the expression of *PCSK7* in either Huh7 or HEK293T cells. * and ** stand for *p* value less than 0.05 and 0.01, respectively.

**Figure 4 metabolites-12-00588-f004:**
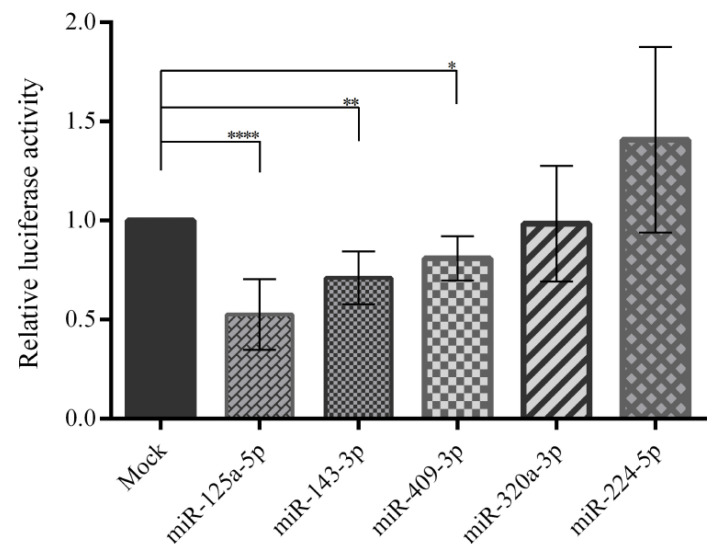
Luciferase assay results following co-transfection of the miR-overexpressing vector and the 3′-UTR of the wild-type *PCSK7* in HEK293T cells. Relative luciferase activity considerably decreased after the overexpression of mir-125a-5p, miR-143-3p, and miR-409-3p; however, there was no significant alteration in relative luciferase activity for miR-320a-3p and miR-224-5p, which were used as controls. *, **, and **** stand for *p* value less than 0.05, 0.01 and 0.0001.

**Figure 5 metabolites-12-00588-f005:**
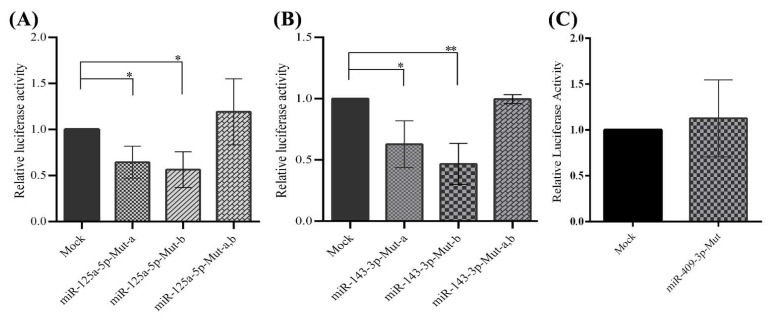
Luciferase assay analysis of cells co-transfected with the miR-overexpressing vector and miRNA target-site-deleted (Mut) psiCHECK-2 vectors in the HEK293T cells. The results demonstrate a direct interaction of miR-125a-5p, miRNA-143-3p, and miR-409-3p with the 3′-UTR of *PCSK7*. (**A**) Both target sites of miR-125a-5p in the 3′-UTR were functional. There was no significant alteration in relative luciferase activity in cells transfected with miR-125a-5p, Mut-a,b in the 3′-UTR of the psiCHECK-2 vector, in which both target sites in miR-125a-5p were omitted. Relative luciferase activity significantly decreased following miR-125a-5p overexpression in Mut-a,b in the 3′-UTR of the psiCHECK-2 vector, in which only one target site of miR-125a-5p was deleted. (**B**) The two miRNA-143-3p target sites in the 3′-UTR of *PCSK7* are functional. No significant alteration was found in the relative luciferase activity in cells transfected with the miR-143-3p-overexpressing vector and Mut-a,b in the 3′-UTR of the psiCHECK-2 vector, in which both miR-143-3p target sites were deleted. However, the relative luciferase activity significantly diminished following the co-transfection of the miR-143-3p-overexpressing vector and Mut-a,b in the 3′-UTR of the psiCHECK vector, in which only one target site was absent. (**C**) No significant changes were observed in luciferase activity after the deletion of the only target site of miR-409-3p in the 3′-UTR of *PCSK7*. Mut-a stands for the deletion of the first target site, Mut-b stands for the deletion of the second target site, and Mut-a,b stands for the deletion of both miRNA target sites. * and ** stand for *p* value less than 0.05 and 0.01, respectively.

**Figure 6 metabolites-12-00588-f006:**
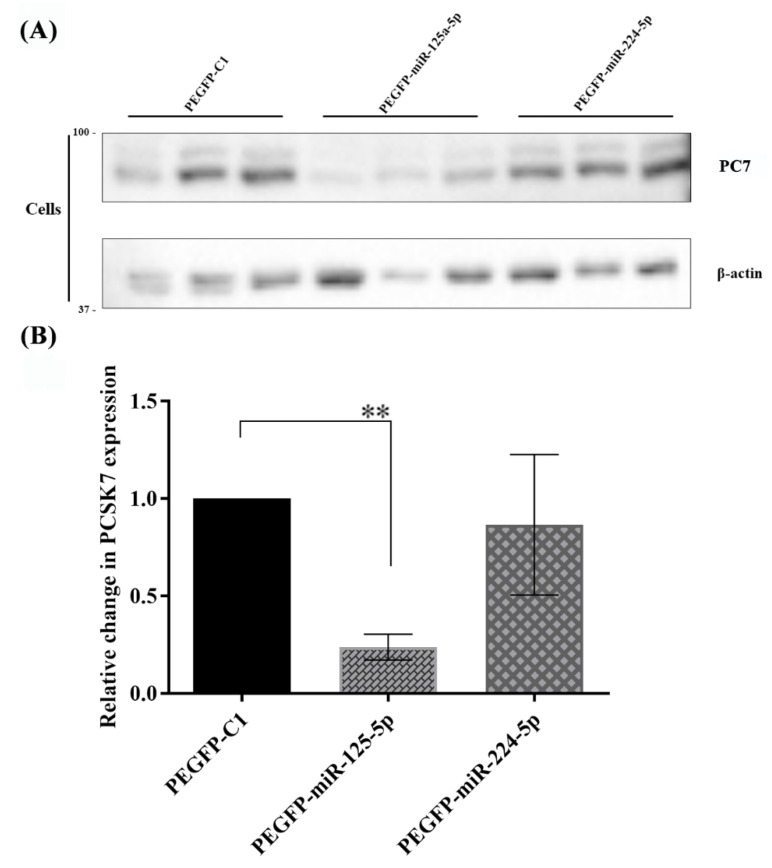
Western blot analysis of PC7 in Huh7 cells, in which miR-125a-5p and miR-224-5p were overexpressed. (**A**) The expression of PC7 protein was significantly diminished by miR-125a-5p at 48 h post-transfection compared to mock counterpart vectors, whereas miR-224-5p did not change the protein levels of PC7. (**B**) The result of quantified bands using the NIH ImageJ software and statistical analysis demonstrated a reduction in the *PCSK7* expression due to the overexpression of miR-125a-5p (*p* = 0.0025) at 48 h post-transfection, whilst there was no significant change in the expression of *PCSK7* following the overexpression of miR-224-5p (*p* > 0.05) in Huh7 cells. ** stand for *p* value less than 0.01.

**Figure 7 metabolites-12-00588-f007:**
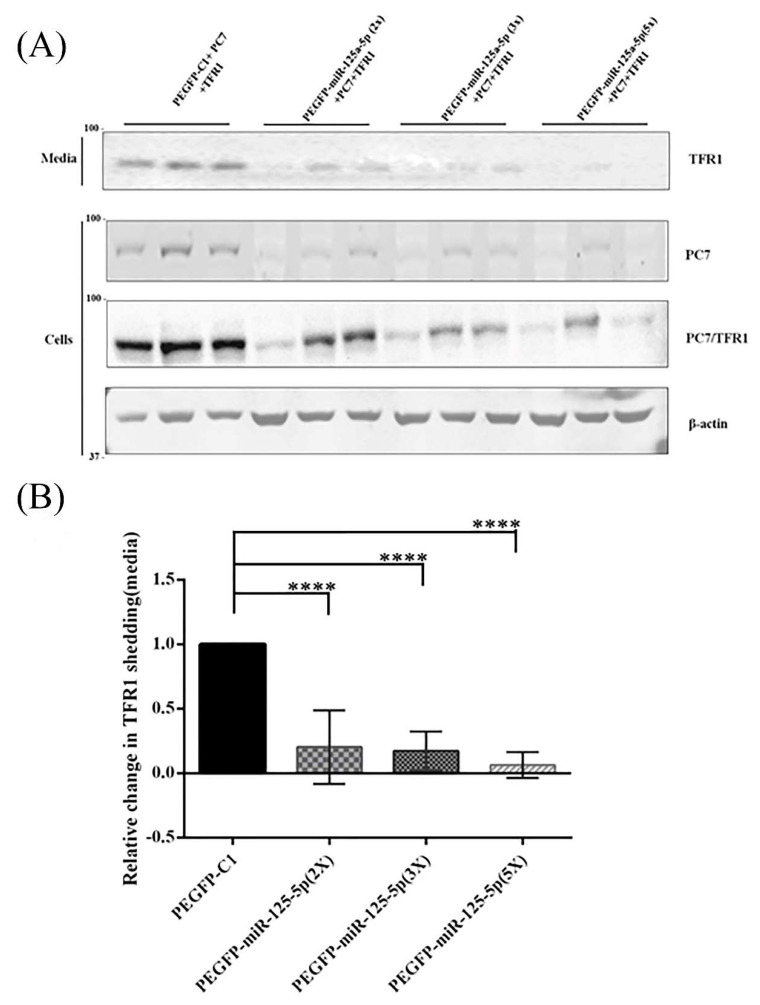
Western blot analysis of the expression of human PC7 and TFR1 after the overexpression of the miR-125-5p vectors in Huh7 cells. The upper panel shows that the levels of shed soluble sTfR1-V5 in the media decreased due to the reduction in PC7 protein levels detected by Western blot in the cells. Here, miR-125a-5p was overexpressed at different plasmid DNA ratios of PCSK7:miRNA: 1:2 (2X), 1:3 (3X), and 1:5 (5X). X stands for vector DNA fold for miR-125-5p compared to vector expressing PC7. Because both PC7-V5 and hTfR1-V5 were V5-tagged, the WB using a mAb-V5 reflects the expression of both proteins. However, WB using a PC7-specific antibody clearly showed that miR-125-5p reduced cellular PC7 protein levels (**A**,**B**), *p* < 0.05, *p* ≤ 0.0001) without any effect on hTfR1 levels (see Appendix A). **** stands for *p* value less than 0.0001.

## Data Availability

Data is contained within the present article or Appendix A.

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
