# Peer review of "Post-Transcriptional Effects of miRNAs on PCSK7 Expression and Function: miR-125a-5p, miR-143-3p, and miR-409-3p as Negative Regulators"

_metabolites, 2022, doi:10.3390/metabo12070588_

Round 1

Reviewer 1 Report

In this study the authors examine post-transcriptional effects of miRNAs on PCSK7 expression and function.

Although the study has the potentiality of being shared with the scientific community, I believe that the manuscript would benefit from a minor revision with the attempt to better support their experimental setting.

1.     The theoretical framework should better describe the scientific evidence that supports the hypothesis they have raised.

2.     Methods section: Experimental procedures should be better defined

3.     I would like to see more of the practical implications. Based on the analyzed variables, how the authors intend to use their findings?

4.     They should include the DOI of all articles to facilitate its later location.

Kind regards

Author Response

Referee 1

  • We have now added a rationale for the studies undertaken and their future practical implications:

End of the Introduction:

Our findings indicate a regulatory network between miRNAs and PCSK7 expression and suggest that the above mentioned three miRNAs are novel modulators of the expression of PCSK7. This information may lead to novel approaches to silence PCSK7 expression in order to mitigate the development of fatty liver disease and its associated sequelae.

End of the Discussion

Overall, we validated our hypothesis that the gene expression of PCSK7 can be downregulated by miRNAs, three of which were identified in this study. In the future, expression of mimics of such miRNAs or others would represent a novel approach to decrease the activity of PC7 and reduce the processing of its downstream substrates, e.g., TfR1, or partners, such as apoA-V and apoB with pharmacological applications in the treatment of fatty liver disease and dyslipidemia.

  • We did our best to better explain the experimental procedures.

  • We added all the DOI to the references as requested.

Regarding the extended applications of our findings, we totally agree with the referee’s suggestion, however, this requires doing additional experiments, which surely will be a future priority. Furthermore, any potential clinical applications of these findings would require much more experiments, including some RNAseq experiments to decipher all transcriptional changes in miRNA treated cells.  Accordingly, we respectfully ask for the indulgence of the referee on this valid point.

Reviewer 2 Report

The manuscript by Malakootian and colleagues described novel miRNAs that target PC7, a member of the proprotein convertases (PCs) family. They further demonstrated a functional outcome for miRNA-mediated regulation of PC7. The data presented support their findings and are novel to the regulation of PC7. I have a few suggestions before the manuscript can be considered for publication. 

1. The resolution for all figures can be improved at later stage.

2. The current layout of figure 2 can be improved. For example, PCSK7 express at 1.5 by QPCR in HEK293T. This data should be better annotated with internal controls. Technically, this is a relative value compared with internal control but an absolute value. It could be misleading when present side-by-side with miRNAs and other cell lines.  

3. In line 271-273, I do not agree with this argument based on the data from figure 2. What is the baseline control used across these three cell lines?

4. Figure 3A, the error bar for miR-320-3p is very large, I am not sure if this data is interpretable or not. What is the n for this data? Can the authors provide a better data?  

5. A typo in line 289. (tow)

6. The quality for Western blots in Figure 7a can be improved.

7. In Figure 7, can the authors present TFR1 blot in the cells? Ideally, intracellular TFR1 level should not decrease because PC7 is reduced by miRNAs. I understand both PC7 and TFR1 are all tagged with V5, hence, it is convenient to use V5 blot to represent both proteins. However, the pattern of the V5 blot (PC7/TFR1) basically only reflects the expression level of PC7. It is hard to tell which band represents PC7 or TFR1. 

Author Response

                                                                                                                          Referee 2

  • We spell checked the whole manuscript and corrected minor errors.
  • We replaced all figures by the best resolution ones we have.
  • Comment #2: With respect to the data in Figure 2, they are all presented as relative expressions, with respect to internal controls (GAPDH for PC7 and RNU48 for microRNAs). That should be independent of cell type.
  • Comment #3: We thank the referee for his/her insightful comment. Despite the complex biological activity of miRNAs, the interaction of miRNAs with their main transcript targets is now well documented for many genes. Nevertheless, in the data presented (Figure 2), we only mentioned what we observed regarding the expression of PCSK7 and our 4 candidate miRNAs. we did not claim that the expression level of PCSK7 is totally and exclusively regulated by these miRNAs. Nevertheless, to resolve any ambiguity we mentioned the following in the discussion of manuscript:

Our study is the first investigation to report regulatory effects of miRNAs on PCSK7 expression and function. We note however that such regulation of PCSK7 must be viewed in the context that each miRNA may target multiple mRNAs and thus regulate complex cellular pathways.  

  • Comment #4: We thank the referee for such an insightful comment. We have performed each experiment as at least two biological replicates (each with three technical replicates included). The variation still existed and could be due to the complex nature of post transcriptional regulation in the cells or in some cases, this could be due to low expression level of miRNA, in which even small change can lead to a large error bar. Despite the reported variation, the data presented remain statistically significant.

  • Comment #5: Thanks for the keen observation, the typo is now corrected.

  • Comment #6: We tried to improve the resolution as best we can.

  • Comment #7: Thank you very much for the comment. We demonstrated that miRNA-12 5a-5p doesn’t target the human TFR1 according to supplementary Figure S4. We are showing in Figure 7 the V5-blot to both hTfR1 and PC7, as requested. However, because of the pandemic this blot is now one year old, and we do not have access to the data anymore to reblot with hTfR1specific antibody in cell lysates. The most important point though is that the secreted shed hTfR1 is enhanced by PC7 expression and that such effect is indeed significantly reduced in the presence of miR-125a.